# Tribological Response of δ-Bi₂O₃ Coatings Deposited by RF Magnetron Sputtering

Sandra E. Rodil [1,*], Osmary Depablos-Rivera [1] and Juan Carlos Sánchez-López [2]

1   Instituto de Investigaciones en Materiales, Universidad Nacional Autónoma de México, Avda. Universidad 3000, Coyoacan, Mexico City 04510, Mexico
2   Instituto de Ciencia de Materiales de Sevilla, Centro Mixto CSIC-Universidad de Sevilla, Avda. Américo Vespucio 49, 41092 Seville, Spain
*   Correspondence: srodil@unam.mx

**Abstract:** Bismuth oxide (Bi₂O₃) coatings and composite coatings containing this oxide have been studied due to their potential applications in gas sensing, optoelectronics, photocatalysis, and even tribology. Two parametric models based on chemical features have been proposed with the aim of predicting the lubricity response of oxides. However, such models predict contradictory values of the coefficient of friction (COF) for Bi₂O₃. In this study, we deposited Bi₂O₃ coatings, via magnetron sputtering, on AISI D2 steel substrates to evaluate the tribological responses of the coatings and determine which parametric model describes them better. Experimentally, only coatings presenting the cubic defective fluorite-like δ-Bi₂O₃ phase could be evaluated. We performed pin-on-disk tests at room temperature and progressively increasing temperatures up to 300 °C using alumina and steel counter-bodies. Low wear and COFs (0.05 to 0.15) indicated that the δ-phase behaves as a lubricious solid, favoring the validity of one of the models. An alternative explanation is proposed for the low COF of the defective fluorite-like structure since it is well known that it contains 25% of anionic vacancies that can be ordered to form low shear-strength planes, similar to the Magnéli phases. Two challenges for future potential applications were observed: one was the low adhesion strength to the substrate, and the other was the thermal stability of this phase.

**Keywords:** lubricious oxide; bismuth oxide; cubic delta-phase; coatings; sputtering; coefficient of friction

## 1. Introduction

The development of coatings to function as solid lubricants for either boundary lubrication conditions or elevated temperatures (above 400 °C) is of special interest to a wide range of industries [1], such as high-speed or dry-cutting machine tools. In boundary lubrication conditions, wear and friction are determined by the interaction between the solid bodies, and heat generation can lead to oxidation. Similarly, most metallic sliding surfaces will oxidize at elevated temperatures [2,3]. Therefore, the oxide films dominate the friction and wear behavior of the interfaces. The most intelligent solution to take advantage of the heating instead of fighting against it is the design of oxide-based lubricant coatings [3–6]. A material qualifies as a solid lubricant when its friction coefficient is less than 0.2, since, at higher COF, the interfaces are largely deformed [7]. A classification of the different solid lubricants is shown in Figure 1 [1,4], where it can be observed that lubrication in high-temperature (HT) applications relies on the use of intrinsic solid lubricants, such as oxides (V₂O₅, AgMo₂O₇), fluorides (CaF₂, BaF₂, CeF₃), hexagonal boron nitride (h-BN) or soft metals (Ag, Cu, Au, Pb, In). The latter are only stable up to 600 °C [1].

Different mechanisms to explain the tribological response of the oxide-based solid lubricants have been envisioned [1], but only the crystal-chemical approaches offered by Erdemir [8] or Prakash and Celis [9] proposed a predictive model to select lubricant oxide surfaces. Among the binary metal oxides, those developing Magnéli phases are particularly

ideal for lubrication at high temperatures [10]. The Magnéli phases are characterized by the organization of oxygen vacancies in such a way that the interlayer distance is increased, leading to the formation of easy shear or crystallographic shear planes [11].

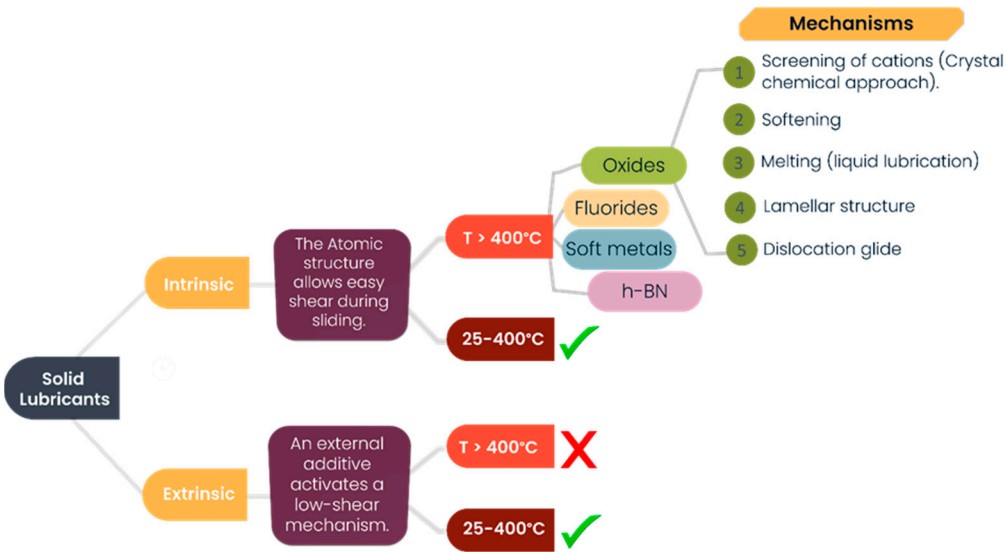

**Figure 1.** Classification of the different solid lubricants.

There is an interest in finding predictors to select lubricious metal oxides since the metallic elements could be added into technological hard coatings (AlCrN, TiSiN, TiAlN), expecting that at high temperatures, the scale is dominated by the lubricious oxide [6], while other mechanical properties are provided by the hard coating. Quantum chemical models have proposed correlations between the lubricity and fundamental properties of the material, such as electronegativity, ionic potential, or chemical hardness [12]. This is so because it has been recognized that the tribochemical or mechanochemical processes during the sliding of two surfaces are initiated and facilitated by the exchange of electrons [13]. Such concepts have been used to explain superlubricity in carbon coatings with and without lubrication [14], but no predictive models have been proposed.

Erdemir [8] showed, using empirical data, an inverse linear correlation between the ionic potential and the coefficient of friction at high temperatures (cf. Figure 2a). The ionic potential (IP = Z/r) is defined as the ratio between the cationic charge (Z) and the radius of the cation (r). The IP is a measure of the screening of the cations by the surrounding anions (oxygen ions). The higher IP indicates that the cations interact strongly with their anions, and there is little interaction with other cations. The reduced cation–cation interaction led to soft oxides, which can shear easily, explaining the trend toward low COF observed in Figure 2a. Nevertheless, Prakash and Celis [9] raised some doubts about the usefulness of the ionic potential, explaining that very ionic oxides, such as BaO, PbO or $Bi_2O_3$, were not included in Erdemir's plot [8,12]. These oxides exhibited low COF (<0.25) and low IP values (1.5, 2, and 2.88 Å$^{-1}$, respectively), contradicting Erdemir's model. According to Prakash and Celis [9], the degree of bonding ionicity is not considered in the IP definition. Alternatively, the authors proposed another marker to consider the interaction between the cation and anion during sliding: the interaction parameter (named *A*) or Yamashita–Kurosawa's interaction parameter [15]. The interaction parameter of an oxide with stoichiometry $M_pO_q$ is a complex function of the polarizability of both the cations and anions (free and bonded) [9].

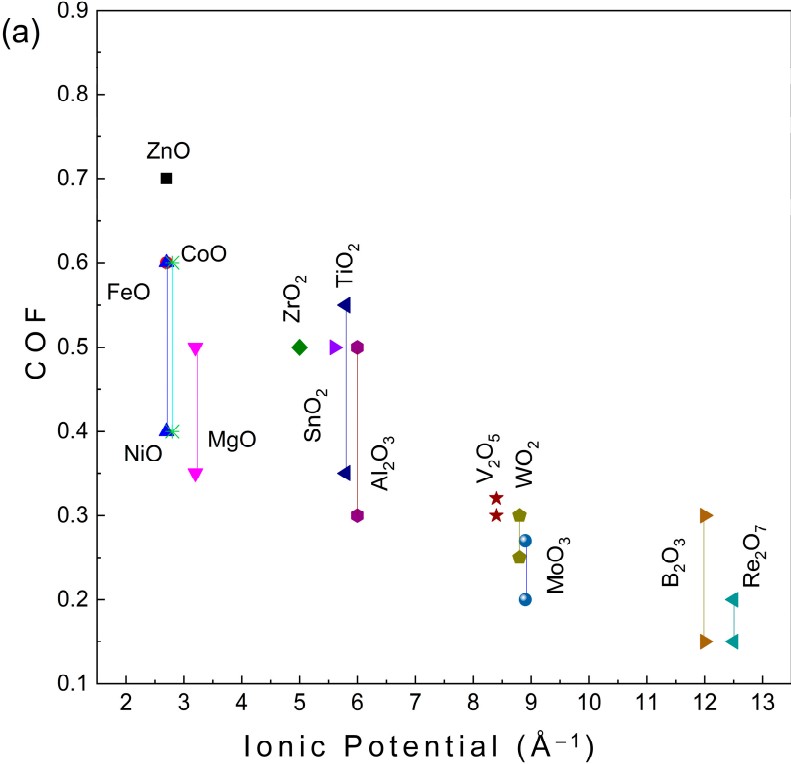

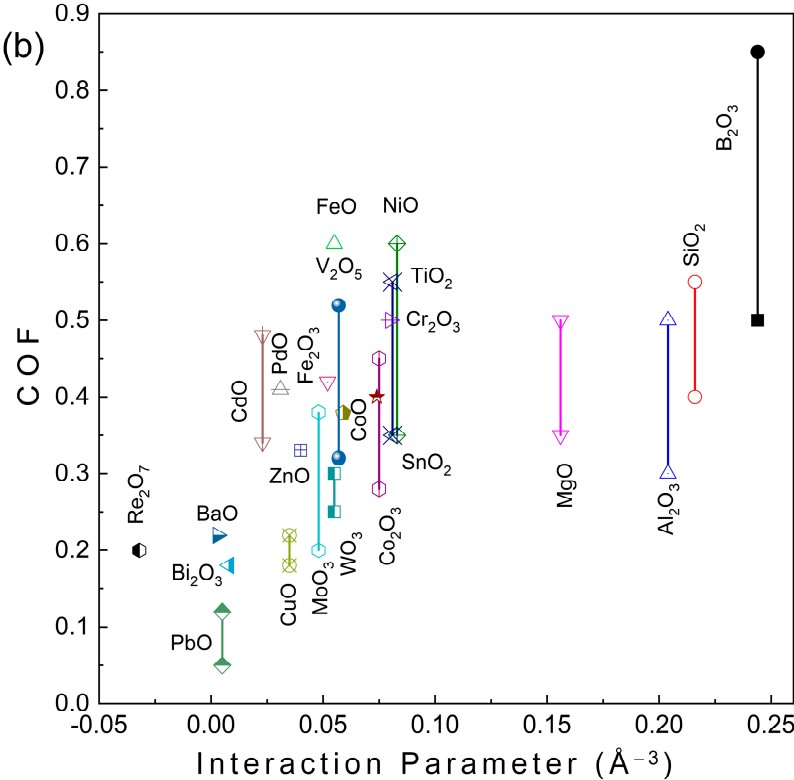

**Figure 2.** Variation in the COF of simple oxides according to (**a**) the ionic potential, reported by Erdemir, Reprinted with permission from Ref. [8]. Copyright 2000, Springer Nature, and (**b**) the interaction parameter, reported by Prakash and Celis, Reprinted with permission from Ref. [9], Copyright 2007, Springer Nature. Vertical lines represent the COF ranges reported in the literature for each oxide under varying tribological test conditions.

The interaction parameter, *A*, is defined as

$$A = \frac{\left[ (\alpha_f^- - \alpha_{O2-}) \right]}{2(\alpha_c + \alpha_f^-)(\alpha_c + \alpha_{O2-})} \tag{1}$$

$\alpha_f^-$ is the polarizability of the anion in the free state, taken as 3.921 Å$^{-3}$ for oxygen based on Pauling values.

$\alpha_{O2-}$ is the polarizability of the anion in the crystal, which, based on Dimitrov and Komatsu calculations [16] for an oxide of stoichiometry $M_pO_q$ (for $Bi_2O_3$, $p = 2$, $q = 3$), can be expressed in terms of the optical bandgap ($E_g$) as

$$\alpha_{O2-} = \left[ \frac{molar\ volume}{2.52} \left( 1 - \sqrt{\frac{E_g}{20}} \right) - p\alpha_c \right] q^{-1} \tag{2}$$

$\alpha_c$ is the cation polarizability, equal to 1.508 Å$^{-3}$ for $Bi^{3+}$ [17].

Using estimated *A* values for different oxides [16] and empirical COFs measured at different temperature ranges (Figure 2b shows the COFs measured at $T/T_m$ intervals between 0.3–1, where $T_m$ is the melting temperature of the oxide), Prakash and Celis [9] predicted that lubricious oxides have COF below 0.2 when their interaction parameter is close to zero, independently of the $T/T_m$ range considered.

Despite the significant differences observed between these two models, not much effort has been made to investigate which chemical parameter could be more appropriate to describe lubricious oxides. The work of Erdemir [8] clearly postulated $Re_2O_7$ and $V_2O_5$ as candidates for low-friction coatings and afterwards, much work was performed to add V into hard coatings to reduce the COF at high temperatures due to the formation of a $V_2O_5$ layer [17–19]. On the other hand, following Prakash and Celis' [9] proposal, some works have evaluated the tribological response of $Bi_2O_3$ added as lubricant nanoparticles [20,21] or into NiAl composite coatings [22].

Bismuth sesquioxide ($Bi_2O_3$) is a very interesting and challenging material that in this context could be used to elucidate the most influencing chemical parameter to predict the lubricious properties of binary oxides, since it has a low ionic potential (2.88 Å$^{-1}$); therefore, a high COF at temperatures close to the melting temperature ($T_m$) is expected according to Erdemir's model [8]. Meanwhile, the empirical model proposed by Prakash and Celis predicts a low COF, since the interaction parameter of $Bi_2O_3$ estimated by Dimitrov and Komatsu [16] at $T/T_m = 0.99$ is very low (0.008 Å$^{-3}$), in agreement with the measured value reported in [9].

However, such models do not consider the dynamic polymorphism of the $Bi_2O_3$ material that presents two thermodynamically stable and many metastable phases [23]. One is the monoclinic ($\alpha$) phase, stable from room temperature (RT) to 730 °C, and the other is the defect-fluorite cubic ($\delta$) phase, which is stable above 730 °C and up to the melting point of the material (830 °C). Two metastable phases ($\beta$-tetragonal and $\gamma$-*bcc*) are formed during the cooling process, and the transformation temperature depends on the cooling rate, ambient atmosphere (vacuum or air), and oxygen concentration. Finally, another two metastable phases have been observed, under very specific conditions ($\omega$ and $\varepsilon$). The $\alpha$-, $\beta$-, $\gamma$- and $\delta$-$Bi_2O_3$ phases have been extensively studied as either bulk materials, nanostructures, or thin films. Thin-film deposition and nanostructure synthesis have resulted in interesting approaches to obtain metastable phases [24], and even the high-temperature $\delta$-phase can be stabilized and retained at ambient conditions, a phenomenon that has been explained as due to the nanometric size of crystals, the film–substrate epitaxial relationship, or the non-equilibrium conditions during the atomic aggregation deposition using sputtering [25–28].

Knowledge about the $Bi_2O_3$ polymorphism is very important, since the data compiled by Prakash and Celis show that $Bi_2O_3$ has a low COF at $T/T_m = 0.99$. At this temperature,

the material has already suffered a phase transition into the δ-$Bi_2O_3$ phase, and although the crystallographic phase is not mentioned, the reported COF must be related to the δ-$Bi_2O_3$ phase. Moreover, the estimation of the interaction parameter, *A*, in Figure 2b for the $Bi_2O_3$ was performed assuming the optical bandgap of the α-phase [16], which is larger than that corresponding to the δ-phase. The implications of these differences are further discussed later.

In this work, we deposited δ-$Bi_2O_3$ coatings (labelled hereafter as δ-BO) by magnetron sputtering to determine their tribological response, which was analyzed in terms of the physico-chemical properties of the bismuth oxide coatings and the lubricious oxide models.

## 2. Materials and Methods

Bismuth oxide coatings were deposited on AISI D2 steel discs using a magnetron sputtering system from an α-$Bi_2O_3$ phase target. According to our previous studies [28,29], we knew that obtention of single-phase $Bi_2O_3$ coatings was only possible for the cubic fluorite-type $Bi_2O_3$ phase, known as the δ phase: radio frequency (RF) power 90 W, substrate temperature 110 °C and deposition time 20 min. The α-$Bi_2O_3$ phase could be obtained after annealing (400 °C for 4 h) of the coatings deposited at 160 W and 200 °C, but as shown later, the phase composition was not uniform. The substrate surfaces were ground using emery papers with different SiC grain sizes and polished to a mirror finish with alumina suspension. This finish promotes a low roughness. Prior to deposition, the substrates were ultrasonically cleaned with isopropanol and dried using hot air.

Coating thickness was confirmed using a Veeco Dektak 150 stylus profilometer. The morphology of the coatings was characterized by field-emission gun scanning electron microscopy (FESEM) using a JEOL 7600F operated at 10 kV. The topography of the samples was measured using a Sensofar S-Neox 090 microscope and a Zygo Nexview TM optical profilometer, and the images were analyzed through Mx TM version 6.4.0.21 software. The film structure was characterized using a Rigaku Ultima IV X-ray diffractometer (Cu Kα radiation, parallel beam optics, measurement mode 2Theta/Omega, step 0.2°, speed 1°/min, measured area 1 cm × 1 cm) and an En-spectra R532 spectrometer for micro-Raman spectroscopy (wavelength 532 nm, magnification 60× and 28.2 mW).

The tribological behavior was studied using two pin-on-disk tribometers from CSM and Microtest (heatable) under different conditions: load (1N or 2N), linear speed (1 cm/s), alumina or steel balls (6- and 10-mm radius) at room temperature and under heating up to 300 °C. Raman measurements of the initial coatings and contact areas after tribological testing were carried out with a LabRAM Horiba Jobin Yvon spectrometer equipped with a charge-coupled device (CCD) detector and a diode-pumped solid-state laser (532 nm) at 5 mW.

## 3. Results

### 3.1. Structural Characterization

The average thickness of the coatings was 1093 ± 34 nm, obtained using the values measured for five different samples deposited under the same conditions.

The coatings presented a uniform globular microstructure with an average grain size of 79 ± 2 nm, as presented in Figure 3a.

The δ-BO coatings were uniform even at a larger scale, as can be observed in Figure 3b, corresponding to the optical profilometry image. The average roughness estimated from 10 measurements was 14.7 ± 0.8 nm, small enough to ensure that the tribological response would not be largely affected. The surface smoothness inherent to the sputtering technique, as observed in the globular surface microstructure, helped to prevent the detachment of debris from protruding particles on the coating, which could act as a third body in the system.

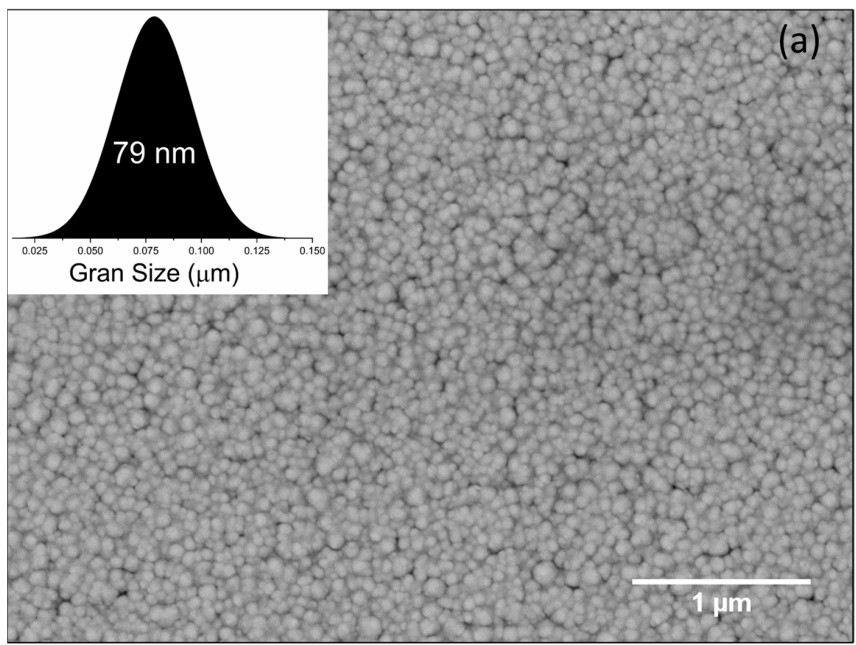

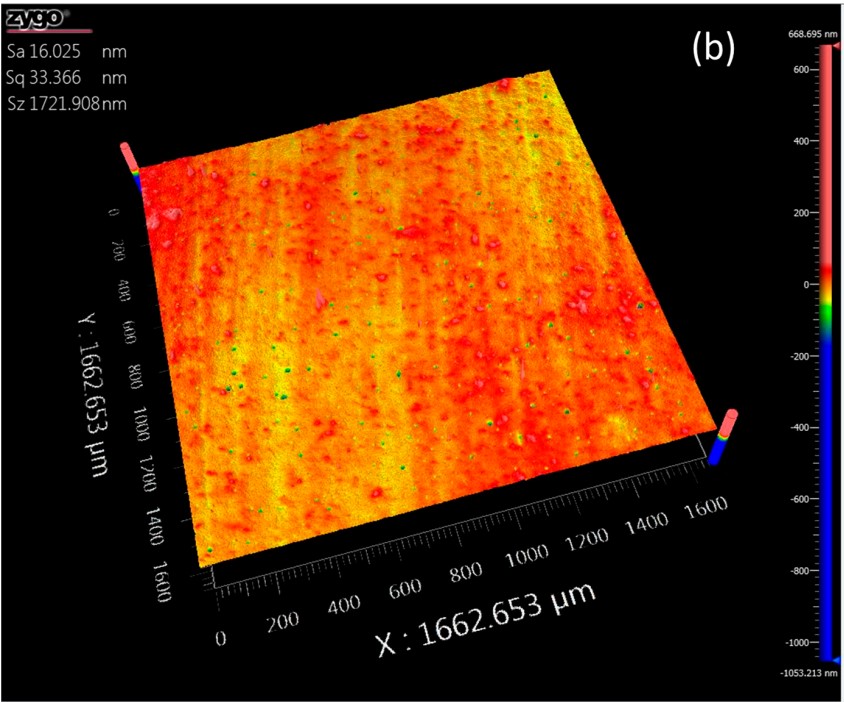

**Figure 3.** Surface of the δ-BO coating deposited on AISI D2 stainless steel substrate. (**a**) Backscattered electron image. The inset shows the grain size distribution obtained using an image analysis software program. (**b**) Representative topographic image obtained by optical profilometry.

The XRD patterns (Figure 4a) measured in different samples presented the cubic phase with the characteristic five peaks of the δ phase. The peak intensity was similar to that reported in the database for a powder sample, i.e., the coatings were polycrystalline without a crystallographic texture. This polycrystallinity agreed with the SEM image (Figure 3a). The additional peak, located at 44.55°, corresponded to that of the steel substrate, which was detected due to the measurement mode used. This is because the penetration depth of the X-rays was longer than the thickness of the coatings. Meanwhile, the micro-Raman spectra (obtained in different spots and samples) showed a single broad symmetric band

around 623 cm$^{-1}$, confirming the disorder-like structure of the δ-Bi$_2$O$_3$ phase (Figure 4b) as described previously [30].

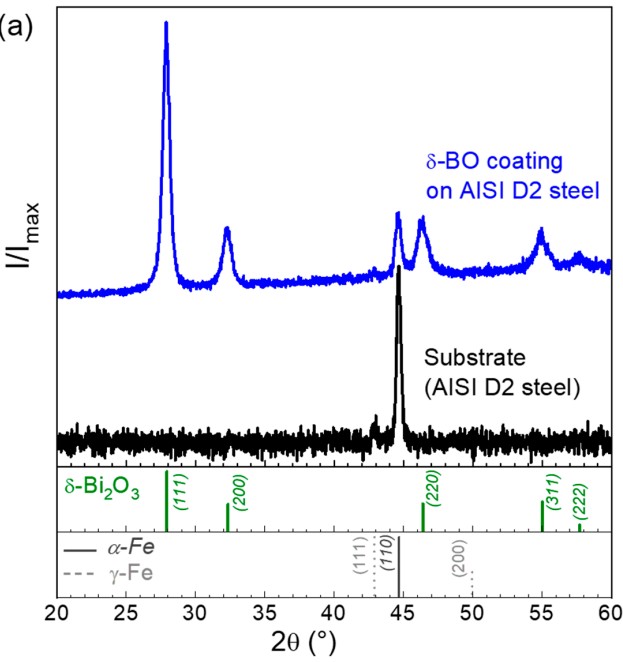

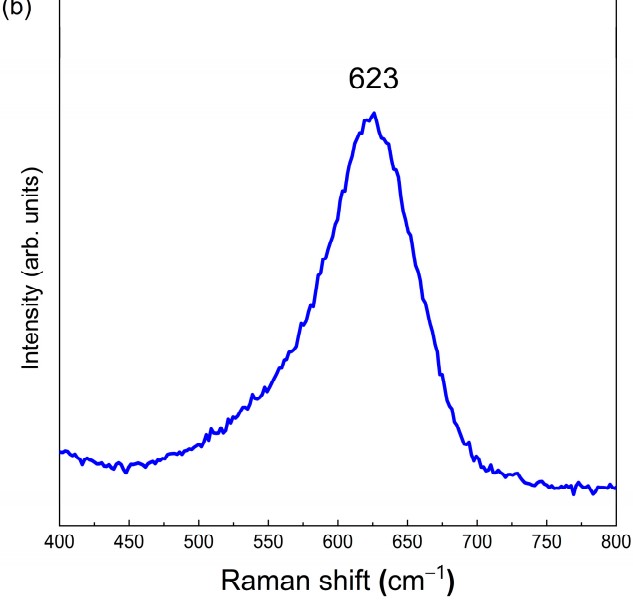

**Figure 4.** Structural characterization that shows the (**a**) XRD pattern and (**b**) Raman spectra of a δ-BO sample. The δ-Bi$_2$O$_3$ phase in the coating, and (α-Fe and γ-Fe) phases from the substrate were identified using the JCPDS cards 00-052-1007, 00-006-0696 and 01-089-4185, respectively.

### 3.2. Tribological Characterization

Figure 5a shows the room temperature coefficients of friction (COFs) obtained for the δ-BO coatings under different loads, counter-bodies and tribometer equipment. It can be observed that steady-state conditions were quickly obtained for both Al$_2$O$_3$ and steel balls during the first 20 cycles. The COF values obtained using the Microtest© equipment varied between 0.08 for the steel balls and 0.15 for the Al$_2$O$_3$ balls. The inset shows the trend of decreasing friction below 0.1 for a longer test (10,000 cycles) under 1 N load using Al$_2$O$_3$ counter-face and the CSM© equipment.

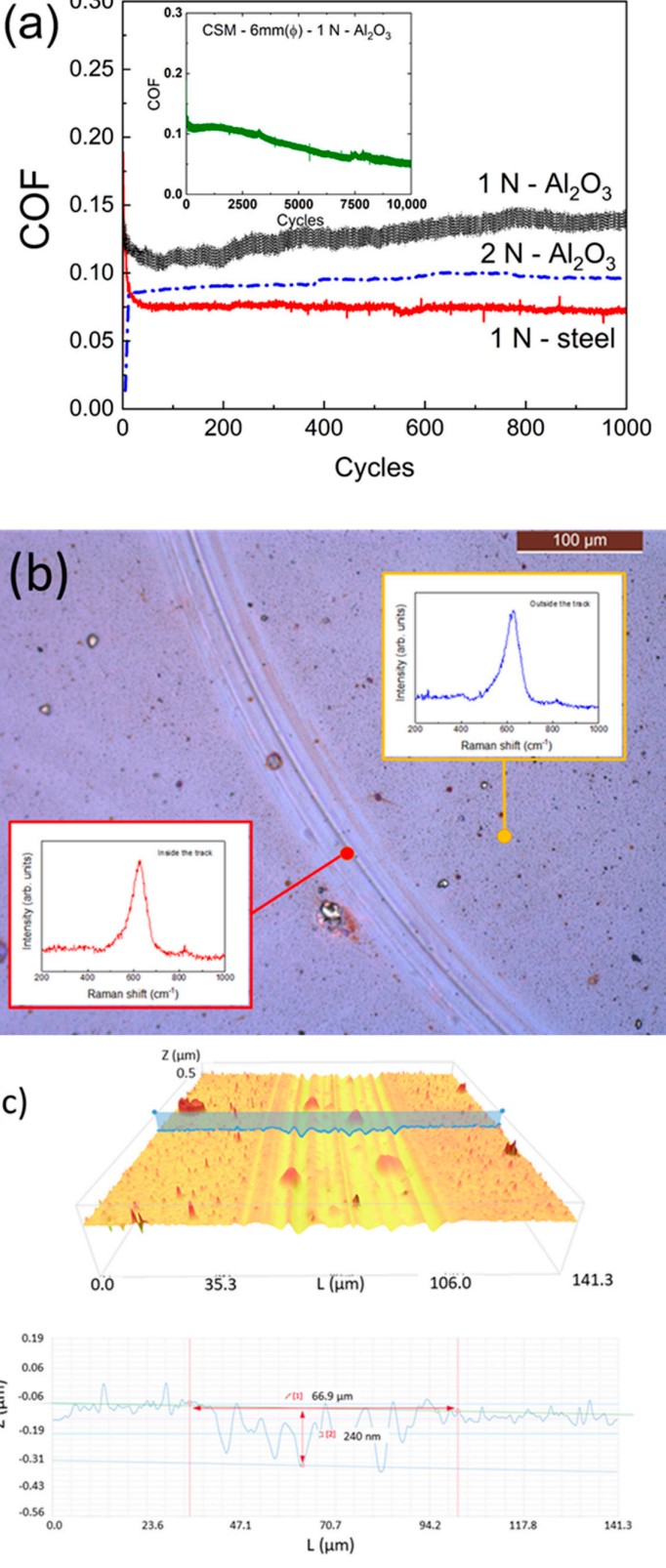

**Figure 5.** Evolution of the coefficient of friction of δ-BO coatings at RT under different testing conditions: (**a**) changing the nature of the mating surface using 10 mm diameter balls. A longer friction test is included in the inset, using 6 mm alumina ball in a CSM tribometer at 1N. (**b**) Optical image of the wear track obtained at room temperature using 1 N of applied load and 6 mm alumina ball, and associated Raman analysis carried out in the initial and rubbed surfaces. (**c**) 3D-topographic image and associated cross-section profile of the wear track shown in the inset of Figure 4a after 10,000 cycles.

When shear forces were applied to the surfaces, the δ-BO coatings showed a tendency to delaminate as a result of a mechanical mismatch, film stresses and/or insufficient chemical bonding. This was manifested by occasionally spontaneous flaking or local spallation inside the wear track (Figure S1 in Supplementary Material). This phenomenon was particularly more pronounced when load was increased (i.e., 2 N instead of 1 N) and observed as a reduction in the number of cycles for which the coefficient of friction remained low (even at RT). The effect of local heating during the RT pin-on-disk tests and the structural stability of the δ-$Bi_2O_3$ phase could not be ruled out, since it has been reported that the δ-BO coatings initiate phase transformation into the tetragonal β-phase at relatively low temperatures (250 °C) [30,31]. The transformation is accompanied by local distortions due to the volume differences among both phases (Figure S2), which might negatively affect the tribological response.

Further tribological analysis to understand the friction mechanism was carried out by studying the wear track for coatings that retained the low COF up to 1000 cycles. Figure 5b exhibits a very smooth wear track, and the Raman spectra obtained inside and outside of the wear track confirm the stability of the δ-$Bi_2O_3$ phase. Figure 5c depicts the 3D topographic image and associated cross-section profile of the wear track obtained after 10,000 cycles (inset Figure 5a), confirming the low wear rate in the steady-state regime at room temperature. No transfer of film to the $Al_2O_3$ ball was observed while the low-friction conditions were retained. Such results support the adhesive failure explanation, which could be improved using adequate bonding layers between the AISI D2 steel substrate and the coating.

The effect of substrate heating on the COF during the pin-on-disk test is presented in Figure 6. As the temperature increased continuously from room temperature to 225 °C, the COF steadily decreased, from 0.15 to 0.05, but above 225 °C, a drastic increase was observed, suggesting a severe modification (Figure 6 and S3).

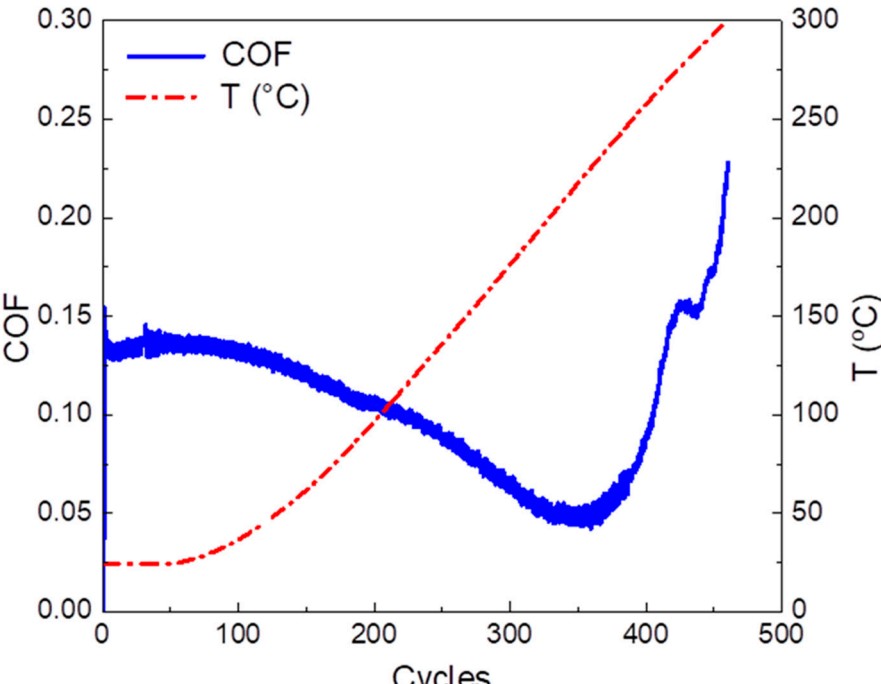

**Figure 6.** Dependence of the coefficient of friction on temperature up to 300 °C using 10-mm alumina ball at 2 N.

Figure 7 depicts a representative set of wear track pictures corresponding to different friction regimes and temperatures. The inner circular track was obtained after a friction test run at room temperature, with low signs of wear corresponding to the low COF

attained (0.09). The following track corresponds to a friction test where the temperature was increasing monotonously from RT to 300 °C, showing clear signs of flaking at different locations, and the friction rose to 0.3 accordingly. The last two wear tracks were obtained from tests carried out after heating the coatings at 300 °C at two different COF values of 0.3 and 0.6, respectively. As noted previously, the coatings tended to break and delaminate easily as the shear and/or temperature increased.

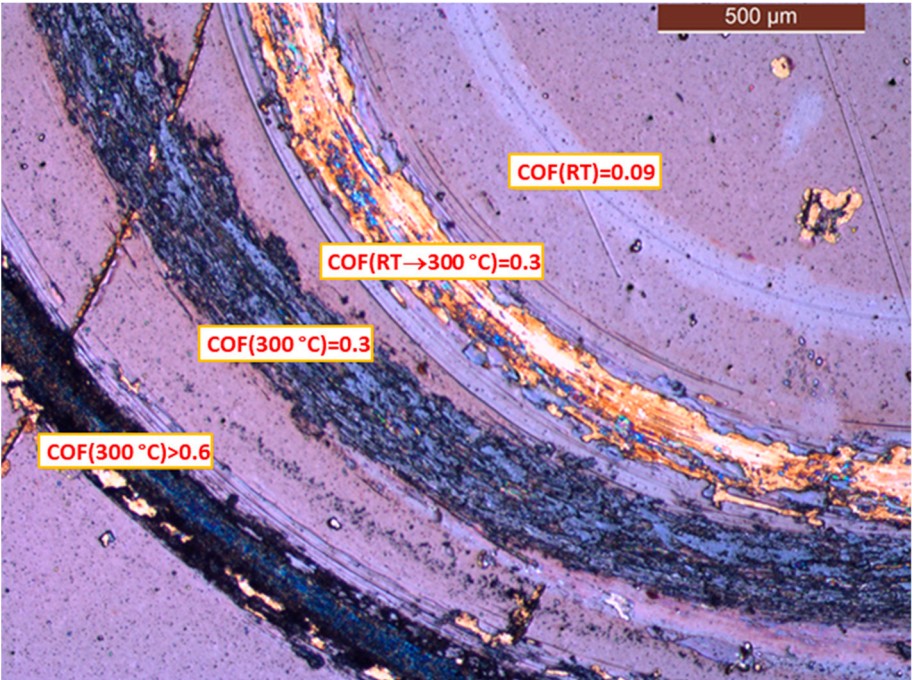

**Figure 7.** Optical pictures of several friction tests carried out at different temperatures and associated friction coefficients. The testing parameters were 2 N of applied load, alumina ball (6 mm in diameter) and linear speed (1 cm/s).

## 4. Discussion

Different tribological systems and conditions were used to evaluate the tribological response of δ-BO coatings deposited on hard AISI D2 steel substrate. The results indicate that the bismuth oxide coatings presenting the δ-$Bi_2O_3$ phase can be considered as a lubricious oxide surface, but for a technological application, much work will be required to achieve adequate coating–substrate adhesion.

The results reported in Figures 5 and 6 clearly demonstrate that the δ-BO coatings present COFs below 0.1, which can be considered of interest for solid lubrication. However, the stability of the cubic-$Bi_2O_3$ phase was limited to about 225 °C, where a phase transition was observed. Figure 6 and S3, corresponding to the pin-on-disk test performed increasing the temperature from RT to 300 °C, show that there is an initial decrease in the COF, which is probably related to a recrystallization of the δ-phase. However, in both measurements, above 225 °C, a sudden COF increase was observed that might be related to the δ-β phase transformation [31]. Raman spectra obtained from the material transferred to the ball after the experiment confirmed the presence of the characteristic peak of the β-$Bi_2O_3$ phase (Figure S3).

The RT tests showed coating–substrate adhesion problems, but when the testing temperature increased over 225 °C, the coating failure was also promoted by the structural transformation associated with the phase change. A similar situation was obtained when coatings annealed after deposition were tested. Annealing was performed with the aim of evaluating the tribological response of $Bi_2O_3$ coatings presenting other crystallographic phases. Deposition and annealing conditions were selected from the structural composition map reported previously [31]. Figure S4 shows the pin-on-disk results for a film presenting

predominantly the $\alpha$-phase. The COF during the first laps was about 0.2 but could not be retained after 20 laps, increasing to values of about 0.6, associated with the film failure. However, the tribological response cannot be assigned univocally to any phase due to the structural inhomogeneity of the annealed coatings (Figures S5 and S6). Moreover, the significant change induced on the surface morphology, affecting the roughness, during the annealing is also evidenced in Figure S7, which could contribute to the increase in the COF.

In terms of the controversy stated at the introduction, the low COFs measured at RT for the $\delta$-BO coatings are close to the COF reported for $Bi_2O_3$ by Prakash and Celis at $T_H$ = 0.99 [9], suggesting that the interaction parameter ($A$ defined in terms of the polarizability of both cations and anions in Equation (1)) is the best predictor for the selection of lubricious oxides based on fundamental chemical properties. Nevertheless, the value of the interaction parameter reported by Prakash and Celis corresponds to an estimation for the $\alpha$-$Bi_2O_3$ phase, since the optical band gap used was 2.8 eV [16]. This is in accordance with measurements and calculations but does not correspond to the high-temperature stable phase of $Bi_2O_3$, i.e., the $\delta$-$Bi_2O_3$ phase. Using an intermediate value of 1.7 eV for the optical bandgap of the $\delta$-phase (reported values vary between 1.5 and 1.8 eV) [28,32], and Equations (1) and (2), we estimated an interaction parameter of $A$ = 0.003 $\text{Å}^{-3}$ for the $\delta$-phase (details shown in the supplementary file). The lower interaction parameter obtained for the $\delta$-$Bi_2O_3$ phase indicates that under the Prakash–Celis model, the COF could be slightly lower than for the $\alpha$-$Bi_2O_3$ phase, which agrees with the values of $\approx$ 0.1 observed for the $\delta$-BO coatings in this work. Moreover, the fact that both interaction parameters fell below 0.2 suggest that $Bi_2O_3$ will present low COFs independently of the crystallographic phase.

The definition of the ionic potential (IP) includes only information about the cation; therefore, parameters related to the structure are not taken into consideration. Accordingly, the IPs of the $\alpha$- and $\delta$-$Bi_2O_3$ phases are the same: 2.88 $\text{Å}^{-1}$. Making a linear regression for the upper COF values in Erdemir's plot (cf. Figure 2a), an estimated value of 0.52 is expected for $Bi_2O_3$ at temperatures close to the melting point, where the $\alpha$-$\delta$ transition should have already occurred [33]. Such a value is much higher than that obtained in this work for the different samples evaluated. However, recent results by Gonzalez–Rodriguez et al. [21], who evaluated the tribological response of different bismuth compounds as lubricants, reported a COF for $Bi_2O_3$ powders ($\alpha$-phase) around 0.45 in the temperature range of RT to 600 °C against steel, close to Erdemir's prediction. Kato and Komai [20] reported that the addition of $Bi_2O_3$ nanoparticles as lubricants lead to a severe-to-mild wear transition when rubbing steel surfaces. Such transition was associated with the formation of a tribofilm promoted by a high oxygen diffusion rate in $Bi_2O_3$, but friction coefficients were not reported.

On the other hand, Sun et al. [22] observed a linear reduction in the COF of Ni-5 wt%Al-X$Bi_2O_3$ composite coatings as the temperature was increased, reaching values between 0.1 to 0.2 at 800 °C when the $Bi_2O_3$ content was increased to a 33 wt.%. They explained the COF reduction in terms of the out-diffusion of the $Bi_2O_3$ to the interphase. Moreover, the lowest COF was related to the $\delta$-$Bi_2O_3$ phase, as demonstrated by thermal analysis. The COF reduction was in accordance with Prakash and Celis' prediction [9].

Direct comparison of those references with our results is difficult due to the completely different tribological systems used, but there seems to be a trend to observe low COFs when the measurements are performed at high temperatures, i.e., where the $\delta$-$Bi_2O_3$ phase predominates.

An important characteristic of the $\delta$-phase is that Bi cations are organized in an *fcc* sublattice, and the O anions are placed in a simple cubic sublattice. The charge stability demands that two O ions are missing per unit cell, and so the structure contains oxygen vacancies that are responsible for the high ionic conductivity of this phase. Various theoretical and experimental studies have been performed to understand the distribution of the vacancies in the O-sublattice [34–36]. Some of these models predict metallic or semi-metallic character for the $\delta$-$Bi_2O_3$ phase, which is not consistent with the experimental

data that reported optical gaps around 1.7 eV. An exception is the model proposed by Matsumoto et al. [35], who assumed that the vacancies are ordered along crystallographic directions (<100>, <111> or <110>), predicting optical gaps between 1.75 and 2.6 eV, in better agreement with experimental observations. Therefore, we wondered if the low COF of $Bi_2O_3$ at temperatures above the $\alpha$-$\delta$ transition might be related to the presence and ordering of the O vacancies, as explained by Gardos et al.: "shear strength (friction) is controlled by anion vacancies" [10].

An alternative explanation related to the tribological response of $Bi_2O_3$ is that the low COF obtained for some samples at high temperatures is related to the presence and aggregation of vacancies to form crystallographic shear planes. Since low COFs are reported at T close to $T_m$, where the $\delta$-$Bi_2O_3$ phase is the thermodynamically stable phase, the high density of vacancies present in the cubic-fluorite structure might play an important role, like in the Magnéli phases.

Magnéli [11], and later others [10,19,37–41], showed that sub-stoichiometric oxides of certain elements (Mo, W, Ti, Re, V) form homologous series of compounds that contain crystallographic shear planes, due to ordering of the oxygen vacancies, with reduced binding strength and, therefore, reduced COF. Bismuth oxides are not catalogued as Magnéli phases, but a recent study by Klinkova et al. [33] concluded that $\delta$-$Bi_2O_3$ is probably the homologous series of $Bi_2O_{3-x}$ phases, whose members have similar structure and differ in their oxygen proportions and the ordering of the oxygen vacancies. Klinkova et al. [33] evaluated, by XRD and electron diffraction experiments, the thermal stability of $Bi_2O_3$ materials under cooling, heating, and different atmospheres to conclude that the distinct phases of $Bi_2O_3$ are oxygen-deficient phases of $Bi_2O_{3-x}$, where $x$ changes according to the temperature range. Such definition is very close to that given to the $TiO_{2-x}$ Magnéli phases [10].

Further studies might be required to demonstrate that $Bi_2O_3$ is a Magnéli oxide, but clearly, the presence of low-shear planes due to the ordering of the anion vacancies could explain the measured low COF and the no-applicability of Erdemir's ionic potential model to the $\delta$-BO coatings. In previous works, we demonstrated the high ionic conductivity of the $\delta$-BO coatings due to the presence of anion vacancies [29]. On the other hand, the polarizability model of Prakash and Celis proposed that oxides with low interaction parameters have low activation energy for the formation of vacancies at the surface. Therefore, increasing the temperature led to an increase in the mobility of the ions due to vacancies that might be the cause of the low COF during sliding. Unfortunately, for the $\delta$-BO coatings, raising the temperature led also to the phase transformation, with the related structural variations that negatively affect the tribological response.

A parallel outcome of this work confirming the lubricity of $\delta$-BO coatings is that it might be interesting to study the addition of small fractions of Bi into well-adhered hard coatings and determine if, during sliding at temperatures close to 730 °C, the $\delta$-$Bi_2O_3$ phase can be formed, reducing the COF, as has been observed with V addition into AlCrN or TiSiN [42–44]. Moreover, metallic Bi could provide lubricity at low temperatures [21,45], and $Bi_2O_3$ at elevated temperatures, to the extent that oxidation occurs.

## 5. Conclusions

The potential use of bismuth oxide coatings deposited on steel substrates presenting the $\delta$-$Bi_2O_3$ cubic phase for solid lubrication has been demonstrated. The as-deposited coatings on mirror-polished AISI D2 steel presented a uniform crystallographic structure and smooth topography, which led to COFs against $Al_2O_3$ or steel balls of about 0.1. Moreover, only a minimal amount of material was transferred to the counter-bodies. However, more work is required to optimize the film–substrate adhesion, since after running at 1 N or 2 N and up to 10,000 cycles, localized spallation areas were observed. Nevertheless, the low friction of the $\delta$-BO coatings was lost when the coating was heated above 225 °C, since, at such temperatures, the structural $\delta$-$\beta$ phase transition initiates with an associated increase in roughness and surface heterogeneity that led to coating failure.

The results presented here suggest that the ionic potential proposed by Erdemir cannot predict the tribological performance of the $Bi_2O_3$ oxides, probably due to the strong polarizability of the Bi-O bonds, or because the existence of oxygen vacancies in the structure has not been considered. The interaction parameter, proposed by Prakash and Celis, appeared as the best predictor of the friction coefficient for $Bi_2O_3$, since it predicts low COFs in accordance with the reported data for the δ-BO coatings. Moreover, the polarizability model is also in agreement with the alternative explanation given for the low COF of $Bi_2O_3$ materials based on the ordering of the intrinsic vacancies to form crystallographic shear planes as observed for the Magnéli-type structures.

**Supplementary Materials:** The following supporting information can be downloaded at: https://www.mdpi.com/article/10.3390/lubricants11050207/s1, Figure S1: Optical observation after finishing the RT pin-on-disk test showing the onset of delamination with local spallation point and surface cracking of the δ-$Bi_2O_3$ where the substrate is observed; Figure S2: Surface micrograph of a δ-$Bi_2O_3$ coating that was heated at 250 °C. Light green areas, indicated with the white arrows, correspond to local distortion zones resulting from the volume differences between the phases involved in the transformation (cubic δ-phase and tetragonal β- phase). The bluish areas correspond to delaminated zones, which were likely caused by the same effect of volume differences; Figure S3: (a) COF measured for a δ-BO sample run against an $Al_2O_3$ ball as the temperature was linearly increased from RT to 300 °C. The COF decreased during the first cycles, and the temperature reached about 250 °C, then it suddenly increased. (b) Optical micrograph of the ball scar and associated Raman spectra of two points characterized by different color (P1-yellow and P2-red zones) can be identified by the light microscope The corresponding Raman spectra looked similar with a single peak at 320 cm$^{-1}$, which cannot be associated to Al2O3. The β-$Bi_2O_3$ phase has a characteristic peak at this position, as reported earlier [1]; Figure S4: Evolution of the friction coefficient of α-BO coating at RT versus the number of cycles: (a) 2500 cycles; (b) detail of first 20 cycles showing the sudden rise of COF, those coatings presented α-$Bi_2O_3$ as the predominant phase; Figure S5: Raman spectra from different zones of a coating deposited by applying 160 W with a substrate temperature of 200 °C, and annealed at 400 °C for 4 h in air to obtain the monoclinic α-$Bi_2O_3$ phase. The circular zones have a different phase (cubic δ-$Bi_2O_3$), which means the no homogeneity of the coating; Figure S6: Grazing angle incidence (1°) XRD pattern of a coating on AISI D2 substrate deposited using the following parameters: rf power applied to the target 160 W, substrate temperature 200 °C, and annealed at 400 °C for 4 h in static air. The identified phases are α-$Bi_2O_3$, δ-$Bi_2O_3$ and the sub stoichiometric oxide $Bi_2O_{2.33}$; Figure S7: Optical profilometry and SEM images of the (a) δ-BO (as-deposited) and (b) films after annealing at 400 °C showing the large differences in the topography induced by the phase transformation reported in Figure S5. These results were obtained for samples deposited on Si substrates simultaneously to the deposition on the AISI D2 substrates.

**Author Contributions:** Conceptualization, S.E.R.; Validation, J.C.S.-L.; Investigation, O.D.-R. and J.C.S.-L.; Writing—original draft, S.E.R.; Writing—review & editing, O.D.-R. and J.C.S.-L.; Funding acquisition, S.E.R. All authors have read and agreed to the published version of the manuscript.

**Funding:** This research was funded by PAPIIT-UNAM Project IV200222.

**Data Availability Statement:** The data presented in this study are available on request from the corresponding author.

**Acknowledgments:** S.E.R. acknowledges technical support from Omar Novelo, Carlos Ramos and Cain Gonzalez.

**Conflicts of Interest:** The authors declare no conflict of interest.

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
