# Peer review of "Tribological Response of δ-Bi2O3 Coatings Deposited by RF Magnetron Sputtering"

_lubricants, doi:10.3390/lubricants11050207_

Round 1

Reviewer 1 Report

Q1

Line 35-36: “The lubricity is due to the facility of the material to deform plastically and/or because they form low-shear-strength surfaces at elevated temperatures [1].”

The plastic deformation of materials, especially non-uniform and severe plastic deformation, can cause severe friction and wear, resulting in the inability to exert lubrication. Please express this sentence rigorously.

Q2

Line 90-94: “Bismuth sesquioxide (Bi2O3) is a very interesting and challenging material that under this context could be used to elucidate the most influencing chemical parameter to predict the lubricious properties since it has a low ionic potential (2.88 â„«-1) and therefore a high COF is expected according to Erdemir’s model, but the Prakash-Celis’s interaction parameter (0.008 â„«-1) predicts a low COF.”

The low friction coefficient of Bi2O3 is closely related to temperature. When predicting the friction coefficient using polarizability or interaction parameters, the specific temperature should be indicated.     

Q3

Line105-106: “and even the high-temperature δ-phase can be quenched and retain at ambient conditions [25-28].”

δ-Bi2O3 is a high-temperature stable phase (729-825 ℃). Please carefully review the evidence proposed in Reference [25-28] for the stable existence of δ-Bi2O3 at ambient temperature.

Q4

Figure 2a: An XRD peak of α-Fe (approximately 44.5°) was detected in δ-BO coating. Please explain the reason. Is it because the detection depth of XRD exceeds the thickness of the coating? Please indicate the size of the area detected by the XRD device used in this experiment.

Q5

Line 155: “When larger loads were applied, the coating suffered adhesive failure. The coatings failure could be due to a low coating-substrate adhesion, which could be improved using adequate bonding layers between the AISI D2 steel substrate and the coating. However, another possible explanation is related to the structural stability of the δ-Bi2O3 phase. In previous works, we reported that the δ-Bi2O3 coatings initiates phase transformation into the tetragonal β-phase at 250°C [29, 30]. The transformation is accompanied by local distortions due to the volume differences among both phases”

This paragraph is all explaining the cause of coating failure, but adhesive failure is not shown in Figure 3a. How many N does 'larger loads' specifically refer to? How to determine if the coating has failed? Does it occur at room temperature?

Q6

Line164-167: “It can be seen that as the temperature increases from room temperature to 225 °C, the COF is steadily reducing from 0.15 to 0.05 but above 225 °C, a drastic increase is observed, suggesting a severe modification. Moreover, optical images indicate that the coating has been delaminated”

Where are the optical images? Please indicate.

Q7

The Si3N4-BO coating friction system in the supplementary file is significantly different from the Al2O3-BO and steel-BO friction systems in the manuscript, and cannot be explained simply by the phrase “These measurements were also confirmed using other system and condition”.

Q8

Under what conditions were each of the four abrasion paths tested in Figure 5, which are not clearly indicated in the figure, and also not specified in the text paragraph, does Figure 5 correspond to the friction coefficients curve in Figure 3? Anyway, it is confusing to read.

Q9

Line 209-210: Experimental data from coatings presenting the α-phase were also obtained, but the films failed after very few cycles.

There are no corresponding pictures or literature to support this expression in the article.

Q10

Apart from room temperature, the crystal forms of Bi2O3 inside and outside the wear trace were not characterized at higher temperatures, such as 100-300 ℃, and there is no persuasive data evidence to prove that the lubrication performance of the coating at RT-200 ℃ is related to the ordered anion vacancies among the δ-Bi2O3. These key data are lacking in the article.

Author Response

Dear Reviewer

Thanks for the detailed revision of the manuscript.

All your comments have been very appropriate and helpful in improving the manuscript.

Q1

 Line 35-36: “The lubricity is due to the facility of the material to deform plastically and/or because they form low-shear-strength surfaces at elevated temperatures [1].”

The plastic deformation of materials, especially non-uniform and severe plastic deformation, can cause severe friction and wear, resulting in the inability to exert lubrication. Please express this sentence rigorously.

R: We have modified the Introduction keeping the message focused on the use of oxides. Therefore, the general statement about lubricity was removed. The introduction now reads as:

 The development of coatings to function as solid lubricants for either boundary lubrication conditions or elevated temperatures (above 500 °C) is of special interest to a wide range of industries [1], such as high-speed or dry-cutting machine tools. In boundary lubrication conditions, wear and friction are determined by the interaction between the solid bodies, and heat generation can lead to oxidation. Similarly, most of the metallic sliding surfaces will oxidize at elevated temperatures [2, 3]. Therefore, the oxide films dominate the friction and wear behavior of the interfaces. The most intelligent solution to take advantage of the heating instead of fighting against it is the design of oxide-based lubricant coatings [3-6]. A material qualifies as a solid lubricant when its friction coefficient is less than 0.2 since, at higher COF, the interfaces are largely deformed [7]. A classification of the different solid lubricants is described in Figure 1 [1, 4], where it can be observed that lubrication at high temperature (HT) applications relies on the use of intrinsic solid lubricants, such as oxides (V2O5, AgMo2O7), fluorides (CaF2, BaF2, CeF3), hexagonal boron nitride(h-BN) or soft metals (Ag, Cu, Au, Pb, In). The latter are only stable up to 600°C [1].

Q2

Line 90-94: “Bismuth sesquioxide (Bi2O3) is a very interesting and challenging material that under this context could be used to elucidate the most influencing chemical parameter to predict the lubricious properties since it has a low ionic potential (2.88 â„«-1) and therefore a high COF is expected according to Erdemir’s model, but the Prakash-Celis’s interaction parameter (0.008 â„«-1) predicts a low COF.”

The low friction coefficient of Bi2O3 is closely related to temperature. When predicting the friction coefficient using polarizability or interaction parameters, the specific temperature should be indicated.    

R: You are right, we must mention the temperature. New text:

Bismuth sesquioxide (Bi2O3) is a very interesting and challenging material that under this context could be used to elucidate the most influencing chemical parameter to predict the lubricious properties of binary oxides since it has a low ionic potential (2.88 ) and therefore a high COF at temperatures close to the melting temperature (Tm) is expected according to Erdemir’s model [8]. Meanwhile, Prakash-Celis’s predicts a low COF since the interaction parameter estimated at T ~ Tm is very low (0.008 ), in agreement with the data reported [9].

Q3

Line105-106: “and even the high-temperature δ-phase can be quenched and retain at ambient conditions [25-28].”

δ-Bi2O3 is a high-temperature stable phase (729-825 ℃). Please carefully review the evidence proposed in Reference [25-28] for the stable existence of δ-Bi2O3 at ambient temperature.

R: We modified the text, replacing the word “quenched” by “stabilized” to avoid confusion, and we mentioned the explanations given for the referenced authors to the stabilization of the δ-Bi2O3 phase at ambient temperature using the different deposition methods.

Modified text:

Thin-film deposition and nanostructure synthesis have resulted in interesting approaches to obtain metastable phases [24] and even the high-temperature d-phase can be stabilized and retained at ambient conditions, phenomena that  has been explained as due to the nanometric size of crystals, the film-substrate epitaxial relationship, or the non-equilibrium conditions during the atomic aggregation deposition using sputtering [25-28].

Q4

Figure 2a: An XRD peak of α-Fe (approximately 44.5°) was detected in δ-BO coating. Please explain the reason. Is it because the detection depth of XRD exceeds the thickness of the coating? Please indicate the size of the area detected by the XRD device used in this experiment.

R:  The peak at ~44.5° corresponding to the substrate is observed because of the measurement mode, which was 2Theta/Omega (Bragg-Brentano). The penetration of X-rays under that mode is about 10 µm, for this reason is inevitable to observe the reflection peaks coming from the substrate in coatings with thicknesses under 10 µm. The intensity is not too high due to the high X-ray absorption by bismuth atoms.

More details about the measurement mode and area were included in the Materials and Methods section, as well as a brief explanation of the observation of the substrate peak at 44.55° was added in the Results section.

Modified text:

The film structure was characterized using a Rigaku Ultima IV X-ray diffractometer (Cu kα radiation, parallel beam optics, measurement mode 2Theta/Omega, step 0.2°, speed 1 °/min, measured area 1 cm x 1 cm)

Modified text:

The XRD patterns (Fig. 4(a)) measured in different samples presented the cubic phase with the characteristic five peaks of the d phase. The peak intensity is similar to those reported in the database for a powder sample, i. e.  the coatings are polycrystalline without a crystallographic texture. This polycrystallinity agrees with the SEM image (Fig. 3(a)). The additional peak located at 44.55° corresponds to the substrate, which was detected due to the measurement mode used. This is because the penetration depth of the X-rays is longer than the thickness of the coatings.

Q5

Line 155: “When larger loads were applied, the coating suffered adhesive failure. The coatings failure could be due to a low coating-substrate adhesion, which could be improved using adequate bonding layers between the AISI D2 steel substrate and the coating. However, another possible explanation is related to the structural stability of the δ-Bi2O3 phase. In previous works, we reported that the δ-Bi2O3 coatings initiates phase transformation into the tetragonal β-phase at 250°C [29, 30]. The transformation is accompanied by local distortions due to the volume differences among both phases”

This paragraph is all explaining the cause of coating failure, but adhesive failure is not shown in Figure 3a. How many N does 'larger loads' specifically refer to? How to determine if the coating has failed? Does it occur at room temperature?

R: Thanks , we have add more information and the description was improved to avoid misunderstanding.

For your revision, we can show you more details.

The coating failure is confirmed by optical observation of the disk where the substrate can be observed at the bottom of the wear track and was later confirmed by measuring the cross-section profile, whose maximum depth is coincident with the film thickness (see enclosed picture).

Local spallation

Coating failure

Cross-section profile

Modified text:

The δ-Bi2O3 coatings showed a tendency to delaminate when applying shear forces to the surfaces as result of a mechanical mismatch, film stresses and/or insufficient chemical bonding. This was manifested by occasionally spontaneous flaking or local spallation inside the wear track (Figure S1). This phenomenon was particularly more pronounced when load is increased (i.e. 2N instead of 1N) and observed as a reduction in the number of cycles for which the coefficient of friction remained low (even at RT). The effect of local heating during the RT pin-on-disk tests and the structural stability of the d-Bi2O3 phase cannot be ruled out since it has been reported that the d-Bi2O3 coatings initiates phase transformation into the tetragonal b-phase at relatively low temperatures (250°C) [30, 31]. The transformation is accompanied by local distortions due to the volume differences among both phases (Fig S2), which might negatively affect the tribological response.

Further tribological analysis to understand the friction mechanism was carried out by studying the wear track for coatings that retained the low COF up to the 1000 cycles.  Figure 5 (b) exhibits a very smooth wear track, and the Raman spectra obtained inside and outside of the wear track confirms the stability of the d-Bi2O3 phase. Figure 5(c) depicts the 3D topographic image and associated cross-section profile of the wear track obtained after 10000 cycles (inset Fig. 5(a)) concluding the low wear rate in the steady-state regime at room temperature. No transfer film to the Al2O3 ball was observed as the low-friction conditions are retained. Such results support the adhesive failure explanation, which could be improved using adequate bonding layers between the AISI D2 steel substrate and the coating. 

Q6

Line164-167: “It can be seen that as the temperature increases from room temperature to 225 °C, the COF is steadily reducing from 0.15 to 0.05 but above 225 °C, a drastic increase is observed, suggesting a severe modification. Moreover, optical images indicate that the coating has been delaminated”

Where are the optical images? Please indicate.

R:  It is certainly confusing as the optical images are not shown. We add an optical image in the supplementary file and modify the text.

Modified text: was included in previous answer.

Q7

The Si3N4-BO coating friction system in the supplementary file is significantly different from the Al2O3-BO and steel-BO friction systems in the manuscript, and cannot be explained simply by the phrase “These measurements were also confirmed using other system and condition”.

R: In sake of clarity, we have removed this information.

Q8

Under what conditions were each of the four abrasion paths tested in Figure 5, which are not clearly indicated in the figure, and also not specified in the text paragraph, does Figure 5 correspond to the friction coefficients curve in Figure 3? Anyway, it is confusing to read.

R: We agree with the referee’s comment. The testing conditions were included in the figure captions but not in the body text. The testing parameters (load, speed, material counterface, etc.) were the same except the temperature. They do not correspond to the tests shown in Figure 3. We therefore clarify these points by including an additional paragraph in the text and modified figure (with one of the labels modified).

Modified text and figure:

Figure 7 depicts a representative set of wear track pictures corresponding to different friction regimes and temperatures. The inner circular track is obtained after a friction test run at room temperature with low signs of wear as correspond to the low COF attained (0.09). The following track corresponds to a friction test where the temperature was increasing monotonously from RT to 300°C showing clear signs of flaking at different locations and the friction is raised to 0.3 accordingly. The last two wear tracks were obtained from tests carried out after heating the films at 300°C at two different COF values of 0.3 and 0.6 respectively. As commented previously, the films tend to break and delaminate easily as the shear and/or temperature are increased.

 Figure 7 Optical pictures of several friction tests carried out at different temperatures and associated friction coefficients. The testing parameters were 2 N of applied load, alumina ball (6 mm of diameter) and linear speed (1 cm/s).

Q9

Line 209-210: Experimental data from coatings presenting the α-phase were also obtained, but the films failed after very few cycles.

There are no corresponding pictures or literature to support this expression in the article.

R: Thanks for calling the attention to this point. We have changed substantially the text.

The new text added in the Discussion section is:

The RT tests showed coating-substrate adhesion problems, but when the testing temperature increased over 225°C the coating failure is also promoted by the structural transformation associated to the phase change.    Similar situation is obtained when films annealed after deposition were tested. Annealing was done aiming to evaluate the tribological response of Bi2O3 films presenting other crystallographic phases. Deposition and annealing conditions were selected from the structural-composition map reported previously [30]. Fig. S4 shows the pin-on-disk results for a film presenting predominantly the a-phase. The COF during the first laps is about 0.2 but could not be retained after 20 laps, increasing to values about 0.6, associated to the film failure.  However, the tribological response cannot be assigned to any phase and is strongly influenced by the structural inhomogeneity of the annealed films (Fig. S5-S6). Moreover, the significant change induced on the surface morphology, including the roughness, during the annealing is also evidence in Fig. S7, which could contribute to the COF rising.

Supplementary figures are:

 Figure S4 Plots of the COF as function of the number of laps resulting to the tribological evaluation of the annealed coatings. Those coatings presented α-Bi2O3 as the predominant phase.

Figure S5. Raman spectra from different zones of a coating annealed at 400 °C for 4 h in air to obtain the monoclinic α-Bi2O3 phase. The circular zones have a different phase (cubic δ-Bi2O3), this means the no-homogeneity of the coating.

Figure S6 Grazing angle incidence (1°) XRD pattern of a coating on AISI D2 substrate annealed at 400 °C for 4 h in static air. The identified phases are α-Bi2O3, g-Bi2O3 and the sub stoichiometric oxide Bi2O2.33.

a)

b)

Sa= 13.992

Sa= 133.7 nm

Figure S7 Optical profilometry and SEM images of the a) δ-Bi2O3 (as-deposited) and b) the 400 °C annealed films showing the large differences in the topography induced by the phase transformation reported in Figure S5.

Q10

Apart from room temperature, the crystal forms of Bi2O3 inside and outside the wear trace were not characterized at higher temperatures, such as 100-300 ℃, and there is no persuasive data evidence to prove that the lubrication performance of the coating at RT-200 ℃ is related to the ordered anion vacancies among the δ-Bi2O3. These key data are lacking in the article.

R: You are absolutely right. We have done many experiments related to the thermal treatment of the Bi2O3 films, so we felt confident that a phase transformation has occurred. Since our tribological system does not have a Raman in situ, we could not obtain the evidence of the phase transformation. After running the different tests, there was extensive degradation in the wear track. The film was gone, so we did not try to perform the Raman analysis.

After your comment, we decided to look at the material transfer to the ball for a test run from RT to 300°C finding evidence of the d-b transformation. The information was added into the supplementary file.

Comment added in the main text (Discussion section):

Raman spectra obtained from the material transferred to the ball after the experiment, confirm the presence of the characteristic peak of the b-Bi2O3 phase (Fig. S3).

Figure S3 a) COF measured for a δ-Bi2O3 sample run against an Al2O3 ball as the temperature was linearly increased from RT to 300 °C. The COF decreased during the first cycles, and the temperature reached about 250 °C, then it suddenly increased. b) Micrograph of the counter-body. The material transfer to the ball is thick, and two colors (yellow and red zones) can be identified by the light microscope. c) Raman spectra corresponding to both color zones of the transferred material, similar spectra were observed for both. There is a single peak at 320 cm-1, which cannot be associated with Al2O3. The β-Bi2O3 phase has a characteristic peak at this position, as reported earlier.

Reviewer 2 Report

The paper "Tribological response of δ-Bi2O3 coatings deposited by RF magnetron sputtering" is suitable for publication in Lubricants after some minor corrections:

1. The abstract is a bit short. The authors should add a first sentence regarding the general use of the Bi2O3 coatings.

2. In general, the introduction is well written, but there are some aspects that need added; the authors should add information about comparisons with other types of coating methods and present the influence of the elements in terms of UTS, YS, hardness, and stiffness, besides COF. Suggested references: 10.3390/coatings10121186, 10.3390/mi12121447.

3. Correlate the XRD data with some microstructure images in the first part of the results.

4. Add an extra paragraph in the conclusion regarding the correlation between COF and the microstructure.

The rest is fine.

Author Response

Dear Reviewer

Thanks for the detailed revision of the manuscript.

All your comments have been very appropriate and helpful in improving the manuscript.

Comments and Suggestions for Authors

The paper "Tribological response of δ-Bi2O3 coatings deposited by RF magnetron sputtering" is suitable for publication in Lubricants after some minor corrections:

  1. The abstract is a bit short. The authors should add a first sentence regarding the general use of the Bi2O3 coatings.

R: We added a few lines mentioning the use of Bi2O3, but the journal suggests 200 words for the abstract.

Modified abstract (196 words):

Bismuth oxide coatings and composite coatings containing this oxide have been studied due to their potential applications in gas sensing, optoelectronics, photocatalysis, and even tribology. Two parametric models based on chemical features have been proposed aiming to predict the lubricity response of oxides. Nevertheless, neither of the models considers the structure. Particularly for Bi2O3, which is polymorphic, contradictory values of the coefficient of friction (COF) were stated.  In this study, we deposited cubic δ-Bi2O3 phase films, via magnetron sputtering, on AISI D2 steel substrates. We performed pin-on-disk tests under room temperature and progressively increasing temperatures up to 300 °C and against alumina and steel counter-bodies. Low wear and COFs (0.05 to 0.15) indicate that the δ-phase behaves as a lubricious solid, favoring the validity of one of the models. An alternative explanation is proposed for the low COF of the defective fluorite-like structure since it is well known that it contains 25% of anionic vacancies that can be ordered to form low shear-strength planes, similar to the Magnéli phases. Two challenges for future potential applications were observed: one is the low adhesion strength to the substrate, and the other is the thermal stability of this phase.

  1. In general, the introduction is well written, but there are some aspects that need added; the authors should add information about comparisons with other types of coating methods and present the influence of the elements in terms of UTS, YS, hardness, and stiffness, besides COF. Suggested references: 10.3390/coatings10121186, 10.3390/mi12121447.

R: In order to keep the objective of the paper clear, we have not included any more complimentary information. The subject is very focused on the tribological response of oxide surfaces.

Moreover, one point requested by the editor is to “(I) Please check that all references are relevant to the contents of the manuscript.”

The suggested references are not relevant to the results presented in the current manuscript.

  1. Correlate the XRD data with some microstructure images in the first part of the results.

R: Thanks for this suggestion. We have added more information about the microstructure of the films.

The films presented a uniform globular microstructure with an average grain size of 79 ± 2 nm, as presented in Fig. 3(a).

Figure 3 Surface of the d-Bi2O3 film deposited on AISI D2 stainless steel substrate. (a) Backscattered electron image. The inset shows the grain size distribution obtained using the Image J software.  (b) Representative topographic image obtained by optical profilometry.

The films are uniform even at a larger scale, as can be observed in Fig. 3(b) corresponding to an optical profilometry image. The average roughness estimated from 10 measurements is 14.7 ± 0.8 nm, small enough to ensure that the tribological response will not be largely affected. A distribution of pores ranging between 2.5 -6.5 mm in diameter can be seen. The surface smoothness inherent to the sputtering technique, as observed in the globular surface microstructure, helps to prevent the detachment of debris from protruding particles on the coating, which could act as a third body in the system.

  1. Add an extra paragraph in the conclusion regarding the correlation between COF and the microstructure.

R: We have added a few more lines in the conclusion to consider the role played by the microstructure. Also, some comments were added in the discussion section.

Modified conclusions:

The potential use of the cubic phase d-Bi2O3 films deposited on steel substrates for solid lubrication has been demonstrated. The as-deposited films on mirror polished AISI D2 steel presented a uniform crystallographic structure and smooth topography, which led to COFs against Al2O3 or steel balls of about 0.1.   Moreover, only a minimal amount of material was transferred to the counter-bodies. However, more work is required to optimize the film-substrate adhesion since after running at 1N or 2N and up to 10,000 cycles, localized spallation areas were observed. The d-Bi2O3 films performed well as solid lubricants reducing friction and wear at temperatures below 225 °C.

Besides the possible application of the Bi-based coatings, the results presented here suggest that the ionic potential proposed by Erdemir cannot predict their tribological performance, probably due to the strong polarizability of the Bi-O bonds, which is not considered in the ionic potential definition, or because the existence of oxygen vacancies in the structure has not been considered. The interaction parameter, proposed by Prakash-Celis appeared as the best predictor of the friction coefficient, but contradictions related to the atomic structure (d or α) raised some doubts. An alternative explanation for the low COF in the cubic d-Bi2O3 coatings is proposed.  The ordering of the intrinsic vacancies present in the d-Bi2O3 phase might create crystallographic shear planes that constitute new slip structures leading to low COF, as observed for the Magnéli type structures.

  1. The rest is fine.

Thanks.

Round 2

Reviewer 1 Report

Q1

“Meanwhile, Prakash-Celis’s predicts a low COF since the interaction parameter estimated at T ~ Tm is very low (0.008 ), in agreement with the data reported [9].”

The interaction parameter 0.008 of δ-Bi2O3 is confirmed at a temperature point, not a temperature range. please carefully check. And this prediction result was not proposed by Prakash Celis. Please refer to “Dimitrov V, Komatsu T. Classification of oxide glasses: A polarizability approach[J]. Journal of Solid State Chemistry, 2005, 178: 831-46.

Q2

The experimental part of the article focuses on the friction coefficient and wear surface characteristics of the deposited δ-Bi2O3 below 300°C. The key data for judging or predicting Bi2O3 lubricity, such as ionic potential, polarizability (interaction parameters) and intrinsic vaccinies, have not been tested or relevant models have been established in this article, so a large number of descriptions about these parameters to predict Bi2O3 lubricity in the discussion and conclusion sections seems unconvincing.

Q3

Line 363-364 “The δ-Bi2O3 coatings performed well as solid lubricants reducing friction and wear at temperatures below 225 °C.”

Is the reason for the lubricity of the δ-Bi2O3 coatings below 225 °C based on polarizability theory? The polarizability theory to predict the lubricity of oxides is closely related to temperature. Below the melting point, the higher the temperature, the smaller the interaction parameter and the better the lubrication. So in the Conclusion, the lubricity of the δ-Bi2O3 coatings at 225 °C is judged by the polarizability (interaction parameter) is problematic.

In addition, please clarify whether the reason for the deterioration of lubricity above 225 °C is mainly related to the deterioration of the bonding strength between the coating and the substrate and the softening of δ-Bi2O3 itself. The data in this article cannot be used to determine the lubricity of δ-Bi2O3 at higher temperatures.

Author Response

Thanks to the reviewer for the comments that make us perform a deeper analysis of the references and our conclusions.

We summarize hereby the comments written by the reviewers and our corresponding reply and associated changes made in the revised version (in blue), if any, beneath each referee’s comment.

Reviewers' comments:

Q1

“Meanwhile, Prakash-Celis’s predicts a low COF since the interaction parameter estimated at T ~ Tm is very low (0.008 ), in agreement with the data reported [9].”

The interaction parameter 0.008 of δ-Bi2O3 is confirmed at a temperature point, not a temperature range. please carefully check. And this prediction result was not proposed by Prakash Celis. Please refer to “Dimitrov V, Komatsu T. Classification of oxide glasses: A polarizability approach[J]. Journal of Solid State Chemistry, 2005, 178: 831-46.

 R: Thanks. Sorry, we mentioned Dimitrov’s work, but the reference was missing. We have improved the phrase to avoid misinterpretations. The specific temperature (T/Tm) value is now stated.

-is the polarizability of the anion in the crystal, which based on Dimitrov and Komatsu calculations [17] for an oxide of stoichiometry MpOq (for Bi2O3, p=2, q=3) can be expressed in terms of the optical bandgap (Eg) as…

Meanwhile, the empirical model proposed by Prakash-Celis’s predicts a low COF since the interaction parameter of Bi2O3 estimated by Dimitrov and Komatsu [17] at T/Tm = 0.99 is very low (0.008 â„«-3), in agreement with the measured value reported in [9].

Q2

The experimental part of the article focuses on the friction coefficient and wear surface characteristics of the deposited δ-Bi2O3 below 300°C. The key data for judging or predicting Bi2O3 lubricity, such as ionic potential, polarizability (interaction parameters) and intrinsic vacancies, have not been tested or relevant models have been established in this article, so a large number of descriptions about these parameters to predict Bi2O3 lubricity in the discussion and conclusion sections seems unconvincing.

 R: We are presenting experimental data related to the tribological response of bismuth oxide coatings presenting the cubic δ-Bi2O3 phase, but certainly not a model. Using data presented and published previously, we tried to understand the tribological behaviour observed in our Bi2O3 coatings. However, it was complicated since the structure of our coatings correspond to the high-temperature phase of the bulk material.

It would have been interesting to test the different Bi2O3 phases, but single-phase coatings are not easy to obtain, and the different cell volumes among the phases create defects that affect the film integrity and substrate adhesion and, thus, the tribological response.

Nevertheless, we think that our results are relevant to open discussion about the subject and evaluate carefully the tribological properties of Bi2O3 materials, which could be very helpful for future applications or just to improve the understanding about the lubricity of oxides.

Q3

Line 363-364 “The δ-Bi2O3 coatings performed well as solid lubricants reducing friction and wear at temperatures below 225 °C.”

Is the reason for the lubricity of the δ-Bi2O3 coatings below 225 °C based on polarizability theory? The polarizability theory to predict the lubricity of oxides is closely related to temperature. Below the melting point, the higher the temperature, the smaller the interaction parameter and the better the lubrication. So in the Conclusion, the lubricity of the δ-Bi2O3 coatings at 225 °C is judged by the polarizability (interaction parameter) is problematic.

R: We are not sure if we understood this comment.

Certainly, our conclusions pointed out that the polarizability model is the best predictor for Bi2O3. However, since our results are related to the δ-BO coatings, which were stabilized at low temperatures due to the nanostructure and/or deposition conditions, we cannot generalize to δ-Bi2O3 materials.  The conclusions were changed consequently to mention only the δ-BO coatings.

 The potential use of bismuth oxide coatings deposited on steel substrates presenting the d-Bi2O3 cubic phase for solid lubrication has been demonstrated. The as-deposited coatings on mirror-polished AISI D2 steel presented a uniform crystallographic structure and smooth topography, which led to COFs against Al2O3 or steel balls of about 0.1.   Moreover, only a minimal amount of material was transferred to the counter-bodies. However, more work is required to optimize the film-substrate adhesion since after running at 1N or 2N and up to 10,000 cycles, localized spallation areas were observed. Nevertheless, the low friction of the d-BO coatings was lost when the coating was heated above 225 °C since, at such temperatures, the structural d - b phase transition initiates with an associated increase in roughness and surface heterogeneity that led to coating failure.

On the other hand, we have added some comments taken from Prakash and Celis’s work related to the role of temperature in the applicability of their model, in the introduction and discussion sections. The big change made, is that we are stating explicitly that the vacancies play a role, but the low COF is not limited to the high-temperature phase of Bi2O3 (δ), since the αphase has also a low interaction parameter and could present low COFs.

:

Introduction

Using estimated A values for different oxides [17] and empirical COFs measured at different temperature ranges (Fig 2 (b) resumes the COFs measured at T/Tm intervals between 0.3 - 1, where Tm is the melting temperature of the oxide), Prakash and Celis [9] predicted that lubricious oxides have COF below 0.2 when their interaction parameter is close to zero independently of the T/Tm range considered.

Discussion

In terms of the controversy stated at the introduction, the low COFs measured at RT for the d-BO coatings is close to the COF reported for Bi2O3 by Prakash and Celis at TH = 0.99 [9], suggesting that the interaction parameter (A defined in terms of the polarizability of both cations and anions in equation (1)) is the best predictor for the selection of lubricious oxides based on fundamental chemical properties. Nevertheless, the value of the interaction parameter reported by Prakash-Celis corresponds to an estimation for the α-Bi2O3 phase since the optical band gap used was 2.8 eV [17], which is in accordance with measurements and calculations, but do not correspond to the high-temperature stable phase of Bi2O3, i. e. the d-Bi2O3 phase. Using an intermediate value of 1.7 eV for the optical bandgap of the d-phase (values reported between 1.5 and 1.8 eV) [29, 33], and equations (1) and (2), we estimated an interaction parameter of A = 0.003 â„«-3 for the d-phase, details shown in the supplementary file. The lower interaction parameter obtained for the d-Bi2O3 phase indicates that under Prakash-Celis model, the COF could be slightly lower than for the α-Bi2O3 phase, which agrees with the values of ≈ 0.1 observed for the d-BO coatings in this work. Moreover, the fact that both interaction parameters fall below 0.2 suggest that Bi2O3 will present low COFs independently of the crystallographic phase.

The definition of the ionic potential (IP) includes only information about the cation, therefore, parameters related to the structure are not taken into consideration. Accordingly, the IPs of α- and d-Bi2O3 phases are the same: 2.88 â„«-1. Making a linear regression for the upper COF values in Erdemir’s plot (cf. Fig. 1(a)), an estimated value of 0.52 is expected for Bi2O3 at temperatures close to the melting point, where the α - d transition should have already occurred [34]. Such value is much higher than the obtained in this work for the different samples evaluated. Though, recent results reported by Gonzalez-Rodriguez et al. [22], who evaluated the tribological response of different bismuth compounds as lubricants, reported a COF for Bi2O3 powders (α-phase) around 0.45 in the temperature range of RT to 600°C against steel, close to Erdemir’s prediction. Kato and Komai [21] reported that the addition of Bi2O3 nanoparticles as lubricants lead to a severe-to-mild wear transition when rubbing steel surfaces. Such transition was associated to the formation of a tribofilm promoted by a high oxygen diffusion rate in Bi2O3, but friction coefficients were not reported.

On the other hand, Sun et al. [23] observed a linear reduction in the COF of Ni-5wt%Al-XBi2O3 composite coatings as the temperature was increased, reaching values between 0.1 to 0.2 at 800°C when the Bi2O3 content was increased to a 33 wt%. They explained the COF reduction in terms of the out-diffusion of the Bi2O3 to the interphase. Moreover, the lowest COF was related to the d-Bi2O3 phase as demonstrated by thermal analysis. The COF reduction was related to the Prakash-Celis’s prediction [9].

Direct comparison of those references to our results is difficult due to the completely different tribological systems used, but there seem to be a trend to observe low COFs when the measurements are done at high temperatures, i. e. where the d-Bi2O3 phase predominates.  

An important characteristic of the d-phase is that the Bi cations form an fcc sublattice and the O anions are placed in a simple cubic sublattice, the charge stability demands that two O ions are missing per unit cell, and so the structure contains oxygen vacancies that are responsible for the high ionic conductivity of this phase. Various theoretical and experimental studies have been done to understand the distribution of the vacancies in the O-sublattice [35-37]. Some of these models predict semi or metallic character for the d-Bi2O3 phase, which is not consistent with the experimental data that reports optical gaps around 1.7 eV.  An exception is the model proposed by Matsumoto et al. [35] who assumed that the vacancies are ordered along crystallographic directions (<100>, <111> or <110>), predicting optical gaps between 1.75 and 2.6 eV, in better agreement with experimental observations. Therefore, we wondered if the low COF of Bi2O3 at temperatures above the α-d transition might be related to the presence and ordering of the O vacancies, as explained by Gardos et al. “shear strength (friction) is controlled by anion vacancies” [10].

Therefore, an alternative explanation related to the tribological response of Bi2O3 is that the low COF obtained for some samples at high temperatures is related to the presence and aggregation of vacancies to form crystallographic shear planes. Since low COF are reported at T close to Tm, where the d-Bi2O3 phase is the thermodynamically stable phase, the high density of vacancies present in the cubic-fluorite structure might play an important role, like in Magnéli phases.

Magnéli [11] and later others [10, 20, 38-42], showed that substoichiometric oxides of certain elements (Mo, W, Ti, Re, V) form homologous series of compounds which contain crystallographic shear planes, due to ordering of the oxygen vacancies, with reduced binding strength and therefore, reduced COF. Bismuth oxides are not catalogued as Magnéli phases, but according to a recent study of Klinkova et al. [34], they concluded that d-Bi2O3 is probably the homologous series of Bi2O3–x phases, whose members have similar structure and differ among them in the oxygen proportion and the ordering of the oxygen-vacancies. Klinkova et al. [34] evaluated, by XRD and electron diffraction experiments, the thermal stability of Bi2O3 materials under cooling, heating, and different atmospheres to conclude that the distinct phases of Bi2O3 are oxygen deficient phases of Bi2O3-x, where x changes according to the temperature range. Such definition is very close to that given to the TiO2-x Magnéli phases [10].

Further studies might be required to demonstrate that Bi2O3 is a Magnéli oxide, but clearly, the presence of low-shear planes due to the ordering of the anion vacancies could explain the measured low COF and the no-applicability of the Erdemir’s ionic potential model to the d-BO coatings. In previous works, we have demonstrated the high ionic conductivity of the d-BO coatings due to the presence of anion vacancies [30].  On the other hand, the polarizability model of Prakash and Celis proposed that oxides with low interaction parameters have low activation energy for the formation of vacancies at the surface. Therefore, increasing the temperature led to an increase in the mobility of the ions due to vacancies which might be the cause of the low COF during sliding. Unfortunately, for the d-BO coatings, raising the temperature led also to the phase transformation with the related structural variations that negatively affect the tribological response.

A parallel outcome of this work confirming the lubricity of d-BO coatings is that it might be interesting to study the addition of small fractions of Bi into well-adhered hard coatings and determine if during sliding at temperatures close to 730 °C, the d-Bi2O3 phase can be formed reducing the COF, as it has been observed with V addition into AlCrN or TiSiN [43-45]. Moreover, metallic Bi could provide lubricity at low temperatures [22, 46], and Bi2O3 at elevated temperatures, as far as oxidation occurred.

Q4

In addition, please clarify whether the reason for the deterioration of lubricity above 225 °C is mainly related to the deterioration of the bonding strength between the coating and the substrate and the softening of δ-Bi2O3 itself. The data in this article cannot be used to determine the lubricity of δ-Bi2Oat higher temperatures.

R: Thanks. We have tried to explain better the two different situations:

  1. . Without heating, the coating failure is mainly related to adhesion failure.
  2. When the measurements are done heating the sample, the main failure mechanism is associated to the d-b phase transition. Experimental evidences were added in the supplementary file.

Nevertheless, it is also known that during sliding, local heating can also induce the phase transformation even for RT tribo-measurements in film spots.. This possibility is thusmentioned although no evidences could be obtained, since after the test, no film was left in the failure zone (as shown in the supplementary file).

The δ-BO coatings showed a tendency to delaminate when applying shear forces to the surfaces as result of a mechanical mismatch, film stresses and/or insufficient chemical bonding. This was manifested by occasionally spontaneous flaking or local spallation inside the wear track (Figure S1). This phenomenon was particularly more pronounced when load is increased (i.e. 2N instead of 1N) and observed as a reduction in the number of cycles for which the coefficient of friction remained low (even at RT). The effect of local heating during the RT pin-on-disk tests and the structural stability of the d-Bi2O3 phase cannot be ruled out since it has been reported that the d-BO coatings initiate phase transformation into the tetragonal b-phase at relatively low temperatures (250°C) [31, 32]. The transformation is accompanied by local distortions due to the volume differences among both phases (Fig S2), which might negatively affect the tribological response.

Round 3

Reviewer 1 Report

The revision of this manuscript and the answers to relevant review questions are reasonable, and we agree to accept it.